# Thermal Inactivation of Different Capripox Virus Isolates

**DOI:** 10.3390/microorganisms8122053

**Published:** 2020-12-21

**Authors:** Janika Wolff, Martin Beer, Bernd Hoffmann

**Affiliations:** Institute of Diagnostic Virology, Friedrich-Loeffler-Institut, 17493 Greifswald-Insel Riems, Germany; janika.wolff@fli.de (J.W.); martin.beer@fli.de (M.B.)

**Keywords:** capripox, LSDV, GTPV, SPPV, lumpy skin disease, goatpox, sheeppox, thermal inactivation, heat inactivation

## Abstract

Capripox viruses (CaPVs) cause a highly contagious poxvirus disease of livestock animals. Working with CaPVs requires laboratories with a high biosecurity level (BSL 3), and reliable inactivation of these viruses is therefore necessary for working in areas or laboratories with a lower biosecurity status. Heat treatment provides a simple and well-established tool for the inactivation due to its substantial advantages (e.g., easy to perform, fast, cheap, and robust). In our study, we determined the time–temperature profiles needed for a fail-safe inactivation procedure using four different CaPV isolates in aqueous solution with and without the addition of protective serum. All four tested CaPV isolates were completely inactivated after 30 min at 56 °C or 10 min at 60 °C. Since different thermal stabilities of other CaPV isolates could not be fully excluded, we recommend an inactivation procedure of 1 h at 56 °C for safe shipment or working in laboratories with lower biosecurity levels than BSL 3.

## 1. Introduction

Capripox viruses (CaPVs), consisting of the three species lumpy skin disease virus, sheeppox virus, and goatpox virus [1], are reported as the cause of the most serious poxvirus diseases of production animals [2,3,4] and are known to affect mainly cattle, sheep, and goats, respectively [5,6]. CaPV outbreaks lead to severe production losses (e.g., by decreased milk production, decreased growth rate and mass loss, temporary or permanent infertility of bulls, and damage to hide and skin), severely affecting national economies as well as the global economy [5,7,8,9,10,11]. Therefore, the World Organization for Animal Health (OIE) classified CaPVs as notifiable diseases [12].

In addition to DNA, proteins, and phospholipids, all poxvirus virions contain carbohydrates, which makes them significantly different from other enveloped viruses. Some characteristics of poxvirus virions are their high resistance to drying and their considerable stability in the environment, remaining infectious even for several months. In comparison to other enveloped viruses, poxvirus virions are less sensitive to organic solvents or disinfectants, and show a high resistance towards different pH values. However, poxvirus virions are highly sensitive to common thermal and chemical sterilization procedures, radiation, and commonly approved disinfection protocols [13].

Members of the Poxviridae family show different sensitivities to thermal treatment [13,14,15,16,17,18]. Only a few publications exist describing the sensitivity of CaPVs to heat treatment. For four different goatpox virus (GTPV) strains, complete inactivation after 60 min at 55 °C, 30 min at 60 °C, and 10 min at 70 °C [18] has been reported, and a titer reduction of approximately 10^4^ tissue culture infectious dose_50_ (TCID_50_) after 60 min at 50 °C has been described for a certain sheeppox virus (SPPV) strain and a certain GTPV strain [17]. In addition, for the serum neutralization test (SNT) and the virus neutralization test (VNT), the OIE recommends heat inactivation of sera for 30 min at 56 °C [19,20]. However, to our knowledge, detailed studies analyzing sensitivity to heat of the three CaPV species are missing.

Experimental studies with infectious CaPVs have to be performed in laboratories with high biosecurity levels, and shipment of samples containing infectious CaPVs requires biosecurity measures to prevent accidental release of the pathogens. Furthermore, experiments performed in an open system (e.g., protein luminescence assays with whole viruses) represent a danger for the contamination of technical devices. In comparison to other methods, thermal inactivation is easy, timesaving, and cost-effective. Due to these reasons, protocols for reliable heat inactivation of CaPVs would be of great benefit.

Here, we determined the time–temperature profile necessary to robustly inactivate different CaPV strains: the lumpy skin disease virus (LSDV) field strain “Macedonia2016” and the LSDV vaccine strain “Neethling” [21] as well as the GTPV field strain “V/103” and the SPPV vaccine strain “V/104” [22]. In addition, a parapox bovis 2 (pseudocowpox (PCPV)) isolate was included in the study, representing another genus of the Poxviridae family. In addition, the protective impact of different sample matrices that are relevant for virus propagation and used in research was investigated.

## 2. Materials and Methods

### 2.1. Virus Preparation and Inactivation Procedure

LSDV field strain “Macedonia2016”, LSDV vaccine strain “Neethling”, and the used PCPV isolate were propagated in MDBK cells (Madin-Darby bovine kidney cells, FLI cell culture collection number CCLV-RIE0261). GTPV field strain “V/103” and SPPV vaccine strain “V/104” were propagated on SFT-R cells (fetal ovine thymus cell line, FLI cell culture collection number CCLV-RIE0043). Each virus preparation was diluted 1:10 in the following three media: phosphate buffered saline (PBS) pH 7.4, cell culture medium with 10% fetal calf serum (FCS) (in the following named ZB), and 100% FCS. Titers (cell culture infectious dose_50_/mL (CCID_50_/mL)) of different approaches are presented in Table 1.

Aliquots of 1 mL were prepared in 2 mL micro tubes (Sarstedt, Nümbrecht, Germany). Heat treatment was performed in a water bath, 4 °C experiments were performed in a 4 °C room. The following time–temperature profiles were used:4 °C: 120 min for all;56 °C: 5, 10, 30, 60, and 120 min for all; 180 and 240 min only for PCPV;60 °C: 5 and 10 min for all; 30, 60, 120, 180 and 240 min only for PCPV.

Subsequently after treatment, samples were frozen at −80 °C. All samples were passaged three times in a 6-well plate format on MDBK and SFT-R cells, using the matching cell culture media containing 10% FCS. Therefore, cells of approximately 80–90% confluence were inoculated with 1 mL of the heat-treated sample and incubated at 37 °C for 7 days in a 5% CO_2_ environment. Subsequently, cells with virus were harvested via freezing at −80 °C and thawing, and another 1 mL was used for the inoculation of the next passage. In addition, three wells of MDBK cells and three wells of SFT-R cells were left uninfected and passaged the same way as negative controls.

### 2.2. DNA Extraction and Real-Time qPCR

Due to the limited sample volume and the decision to use 1 mL for the passaging of samples in cell culture, 10 µl of each sample was diluted in 90 µl of the respective medium (PBS, ZB, or FCS) to receive 100 µl for the extraction of DNA. Nucleic acid extraction was done using the NucleoMag Vet kit (Macherey-Nagel, Düren, Germany) according to the manufacturer’s instructions on the KingFisher Flex System (Thermo Scientific, Darmstadt, Germany), with some modifications in the volume of certain substances [22]. For the four CaPV isolates, the viral genome loads were determined using a pan-capripox real-time qPCR amplifying a region in the p-32 gene [2] in combination with a modified probe [23]. PCPV genomes were detected using a real-time qPCR assay targeting the B2L gene [24]. An internal control DNA (IC2-DNA) was added for the control of successful DNA extraction [25]. For both used real-time qPCR assays, analytic sensitivity of less than 10 copy numbers per PCR reaction are reported [2,24].

## 3. Results

The effect of thermal treatment could be examined nicely via analyzing the Cq-values of specific real-time qPCR assays for all five tested virus isolates following cell culture inoculation and passaging (Appendix A). This procedure was chosen to ensure that even the smallest amounts of replicable virus could be detected due to the three passages performed. In addition, there was no specific antibody available for pseudocowpox virus, which is why detection via immunofluorescence was replaced by real-time qPCR analyses. Since uninfected cell culture controls remained negative during the whole study in all cases (Appendix A), contamination between the different samples can be excluded. Cq-values directly after thermal treatment ranged between 20.5 (LSDV-“Neethling” vaccine strain) and 26.5 (PCPV isolate). After complete thermal inactivation confirmed by cell culture inoculation, clearly higher Cq-values could be observed over the three cell culture passages. In contrast, samples that were not completely inactivated by heat treatment showed Cq-values that were similar or lower compared to the starting materials (Appendix A). All four different CaPV isolates, independent from the strain type (vaccine or field strain), showed very similar results regarding the heat treatment for different time periods in the different background matrices, whereas clear differences could be observed for the PCPV isolate (Figure 1, Appendix A).

All four CaPV isolates were fully inactivated after 30 min at 56 °C or 10 min at 60 °C, independent of the dilution medium used (Figure 1, Appendix A). In contrast, the first successful inactivation of PCPV at 56 °C could be observed after 2 h in two of three sample matrices (PBS and ZB). Moreover, 10 min at 60 °C was also not sufficient for inactivation of the used PCPV isolate. Here, 180 min at 56 °C or 30 min at 60 °C were necessary for the reliable inactivation of all replicates in the different sample matrices (Figure 1, Appendix A).

## 4. Discussion

For a safe and cheap shipment of samples containing CaPVs, the testing of infectious samples in open systems using technical devices, and for experiments in laboratories with a lower biosecurity level (BSL) than BSL 3, reliable inactivation of CaPVs has to be ensured. Thermal inactivation is characterized by low costs, easy performance, high repeatability and robustness, and time effectiveness. Semi-quantitative analysis of inactivation using specific real-time qPCR assays proved to be an easy, quick, and reliable method. Successfully inactivated samples displayed a clear increase in Cq-value, whereas Cq-values of non- or partially inactivated samples were similar of lower compared to the starting material. Cell culture-adapted capripox virus isolates grow very robustly on their respective cell lines. After 7 days of incubation, as it was performed in the presented study, approximately 80–90% cytopathic effect can be observed. Inoculation volume does not seem to play a major role for the efficacy of virus growth when the virus isolate is adapted on the cell line. Therefore, Cq-values comparable to those of the starting material are not unexpected. Cq-values lower than the starting material can be explained by the preceding 1:10 dilution of the viruses in the different media, which normally means a decrease of approximately 3.3 Cq-values.

For performance of the SNT and the VNT, the OIE proposes thermal inactivation of CaPVs at 56 °C for 30 min [19,20]. We were able to confirm this recommendation. In our study, all four CaPV isolates were completely inactivated after 30 min incubation at 56 °C or 10 min at 60 °C. Since some approaches in our study were already inactivated after 5 min at 60 °C, whereas others still contained some infectious virus (Figure 1, Appendix A), this time–temperature profile (5 min, 60 °C) seems to be the limiting case for CaPVs. CaPV isolates analyzed in the present study displayed inactivation after shorter treatment times when compared to previous descriptions. Kavitha et al. reported, for example, inactivation of different GTPV strains after 60 min at 55 °C and 30 min at 60 °C [18]. It cannot be excluded that the existing difference of 1 °C between the study of Kavitha et al. and our study has an influence on the inactivation procedure. Furthermore, some differences in the thermal stability of different CaPV isolates cannot be fully excluded.

However, as expected, longer time periods were necessary for a complete and robust thermal inactivation of the analyzed parapox virus isolate. Here, 2 h to 3 h at 56 °C or 30 min at 60 °C were necessary (Figure 1, Appendix A), which fits nicely with the data of Rheinbaben et al., who report 2.5 h at 56 °C for a successful inactivation procedure [13]. In general, the impact of the matrices used for dilution could neither be confirmed nor refused in the present study. However, a possible protective effect of FCS could be assumed due to the recorded data; however, marked differences between the used media could not be observed (Figure 1). In conclusion, complete inactivation of CaPVs could be achieved after 30 min at 56 °C, which is suitable for SNT, VNT, and experiments performed in BSL 3-laboratories. However, considering the obtained data and results reported in literature, for safe shipment of CaPV or working in BSL 1 or BSL 2 laboratories, we suggest a prolonged inactivation procedure of CaPVs of 1 h at 56 °C.

## Figures and Tables

**Figure 1 microorganisms-08-02053-f001:**
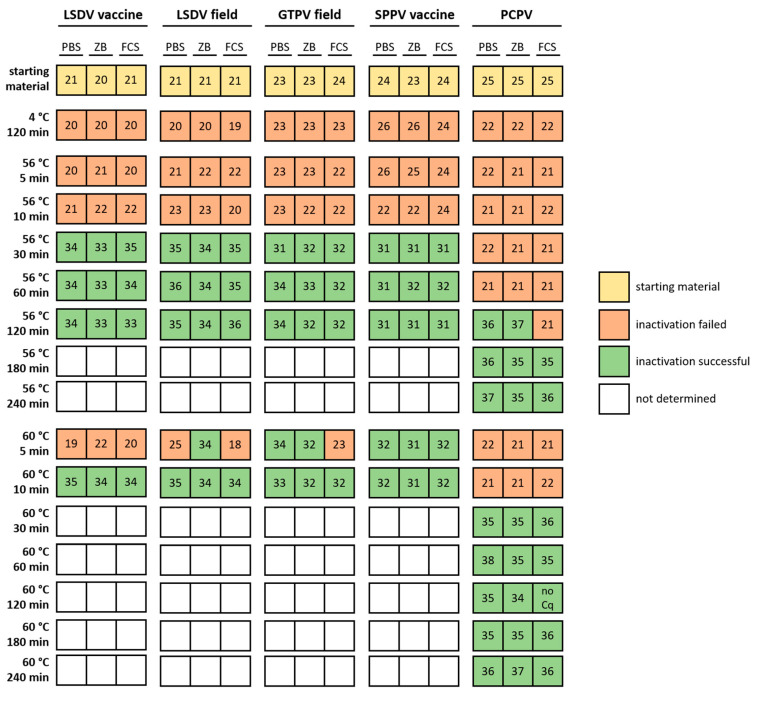
Thermal inactivation of different capripox virus strains as well as a pseudocowpox virus isolate. Four different capripox virus strains (LSDV vaccine strain “Neethling”, LSDV field strain “Macedonia2016”, GTPV field strain “V/103”, and SPPV vaccine strain “V/104”) and a single pseudocowpox virus isolate (PCPV) were diluted in different media (phosphate buffered saline pH 7.4 (PBS), cell culture medium with 10% FCS (ZB), and FCS only) and treated with different temperatures for different periods of time. Inactivation was validated or refused via analysis of the viral genome loads over three cell culture passages. Numbers indicate Cq-values. For the starting material, average Cq-value is given, and for thermal treatment, the Cq-value of the third cell culture passage is presented. Due to the reported analytical sensitivity of both assays of detecting less than 10 copy numbers/PCR reaction, a Cq of around 37 can be equated with approximately 10 copy no./PCR reaction. Therefore, a Cq of around 27 displays approximately 10,000 copy no./PCR reaction, a Cq of around 23 stands for approximately 100,000 copy no./PCR reaction, and a Cq of around 20 is related to approximately 1,000,000 copy no./PCR reaction.

**Table 1 microorganisms-08-02053-t001:** Virus titer of material used for thermal inactivation studies. LSDV: lumpy skin disease virus; GTPV: goatpox virus; SPPV: sheeppox virus; PCPV: pseudocowpox; CCID_50_: cell culture infectious dose_50_; PBS: phosphate buffered saline; ZB: cell culture medium with 10% fetal calf serum; FCS: fetal calf serum.

Virus Isolate	CCID_50_/mL		
	PBS	ZB	FCS
LSDV vaccine strain “Neethling”	10^6.1^	10^6.1^	10^5.9^
LSDV field strain “Macedonia2016”	10^6.9^	10^5.7^	10^5.9^
GTPV field strain “V/103”	10^5.5^	10^5.9^	10^5.1^
SPPV vaccine strain “V/104”	10^5.3^	10^5.9^	10^5.5^
PCPV	10^6.3^	10^5.3^	10^5.1^

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
