# Peer review of "Thermal Inactivation of Different Capripox Virus Isolates"

_microorganisms, 2020, doi:10.3390/microorganisms8122053_

Round 1

Reviewer 1 Report

The authors focused in this manuscript on the determination the time-temperature profiles necessary to inactivate four different Capripox viruses. The work is interesting. Only my doubt and remark, that I want the authors to explain, concerns samples that were not completely inactivated by heat treatment. I dont understand why these samples had similar or lower CT values compared with starting materials. If the virus were not completely inactivated, should it multiply faster or am I wrong? This issue has not been raised in the discussion but should be.

The abstract should be a single paragraph and should follow the style of structured abstracts 1) Background 2) Methods 3) Results and 4) Conclusion. Please correct.

Author Response

The authors focused in this manuscript on the determination the time-temperature profiles necessary to inactivate four different Capripox viruses. The work is interesting. Only my doubt and remark, that I want the authors to explain, concerns samples that were not completely inactivated by heat treatment. I dont understand why these samples had similar or lower CT values compared with starting materials. If the virus were not completely inactivated, should it multiply faster or am I wrong? This issue has not been raised in the discussion but should be.

We thank the reviewer for carefully reading the manuscript and spending interesting thoughts and corrections. Cell culture-adapted capripox virus isolates grow very robustly on their respective cell lines. After 7 days of incubation, as it was performed in the presented study, approx. 80-90% cytopathic effect can be observed. Inoculation volume does not seem to play a major role for efficacy of virus growth, when the virus isolate is adapted on the cell line. Therefore, Cq-values comparable to those of the starting material are not unexpected. Cq-values lower than the starting material can be explained by the preceding 1:10 dilution of the viruses in the different media, which normally means a decrease of approx. 3.3 Cq-values. This information was added to the discussion part of the manuscript.

The abstract should be a single paragraph and should follow the style of structured abstracts 1) Background 2) Methods 3) Results and 4) Conclusion. Please correct. 

Abstract was revised and results, discussion and conclusion were added.

Reviewer 2 Report

Experimental design: authors did the tests based on the assumption of "contact or skin abrasion". From pathogenesis of point view, capripox also transmits by respiratory route (aerosol) as they are particularly resistance to drying (mentioned in line 30). Authors should consider testing the wipes of tube's cap after heat treatment and the wipes of  lab bench or hood, or environment etc.

Abstract: authors should include some numerical data such as 56 C for 30 min or 60 C for 10 min can successfully inactivate the capripox in general.

Lines 51-58: it belongs to the materials and methods.

Table 1: the titers of  tested samples ranged 105-6 (I love to see what  the unit is, pfu or TCID or something else) which is a reasonable range.  I am curious if authors have tested 107-8, although I am not sure whether you can grow capripox to these high titers in cell culture, I would like to know if higher titers would affect the conclusion of data presented Fig. 1.

Fig. 1: the presentation is concise; but this is qualitative, which are converted from quantitative data (Ct values) to which authors refer to supplement files. I would like to see the Ct values ranges of  inactivation failed or inactivation successful for each virus, at least in the figure legend.

Discussion: Authors compare their data to Kavitha et al., and Rheinbaben et al., What are the titers they begin with and are these titers account for the discrepancy or agreement of their data compare with yours.

Author Response

Experimental design: authors did the tests based on the assumption of "contact or skin abrasion". From pathogenesis of point view, capripox also transmits by respiratory route (aerosol) as they are particularly resistance to drying (mentioned in line 30). Authors should consider testing the wipes of tube's cap after heat treatment and the wipes of lab bench or hood, or environment etc.

We thank the reviewer for the assessment of our work and corrected the manuscript in accordance with the reviewer’s comments. The study was not designed to determine the thermal stability of abrasions or contact transmission, but inactivation procedure of virus in aqueous phase. Examination of thermal stability of dried virus originating from animal samples or from wipes of hood are indeed a very interesting question, but would go beyond the scope of the present manuscript. However, we fully agree with the reviewer that additional experiments should be performed in the future to further investigate different additional approaches.

Abstract: authors should include some numerical data such as 56 C for 30 min or 60 C for 10 min can successfully inactivate the capripox in general.

Results in form of numerical data were added to the abstract section.

Lines 51-58: it belongs to the materials and methods. 

We shortened this part in order to avoid mentioning material and methods in detail. Now, only the aim of the study and the used virus isolates are named in this part of the manuscript.

Table 1: the titers of tested samples ranged 105-6 (I love to see what the unit is, pfu or TCID or something else) which is a reasonable range. I am curious if authors have tested 107-8, although I am not sure whether you can grow capripox to these high titers in cell culture, I would like to know if higher titers would affect the conclusion of data presented Fig. 1.

Titers are given in CCID50/ml. We clarified this in the manuscript. The ask for reliable inactivation of virus with higher titer is indeed an interesting question. However, cell culture-adapted capripox viruses normally grow in cell culture with titers around 106/107 CCID50/ml. Due to the different sample matrices used in the presented study to determine a possible effect of protective serum, cell culture virus was diluted in the respective media. Thereby, the titer decreased approx. one log10. For experiments with higher titers, cell culture virus has to be concentrated first and then re-diluted in the respective media. Since 10x concentration does not automatically result in increase of titer of approx. one log10 and because this would have meant additional manipulation of virus, we decided against it.

Fig. 1: the presentation is concise; but this is qualitative, which are converted from quantitative data (Ct values) to which authors refer to supplement files. I would like to see the Ct values ranges of inactivation failed or inactivation successful for each virus, at least in the figure legend. 

Average Cq value of starting material as well as Cq values of third cell culture passage after thermal treatment were included into Figure 1.

Discussion: Authors compare their data to Kavitha et al., and Rheinbaben et al., What are the titers they begin with and are these titers account for the discrepancy or agreement of their data compare with yours. 

Unfortunately, titers are not mentioned in the references which makes the data difficult to compare. Due to the fact that information about thermal inactivation of capripox viruses is highly limited, we decided to cite these references anyway.

Round 2

Reviewer 2 Report

In section 2.2 and Figure 1, authors have mentioned several references (2, 23, 24, 25)  on real-time qPCR applied to 5 isolates.  Is there a way to express the so called "analytical sensitivity" based on Cq-values, as those in traditional way, for example, copy number, based on the Cq-values.  If yes, the analytical sensitivity for each isolate should be added in Figure 1.
